# Development in Maillard Reaction and Dehydroalanine Pathway Markers during Storage of UHT Milk Representing Differences in Casein Micelle Size and Sedimentation

**DOI:** 10.3390/foods11101525

**Published:** 2022-05-23

**Authors:** Miguel Aguilera-Toro, Nina Aagaard Poulsen, Marije Akkerman, Valentin Rauh, Lotte Bach Larsen, Søren Drud-Heydary Nielsen

**Affiliations:** 1Department of Food Science, Aarhus University, Agro Food Park 48, 8200 Aarhus, Denmark; miguel.aguilera@food.au.dk (M.A.-T.); nina.poulsen@food.au.dk (N.A.P.); maray@arlafoods.com (M.A.); lbl@food.au.dk (L.B.L.); 2CiFOOD Centre, Aarhus University, Agro Food Park 48, 8200 Aarhus, Denmark; 3Arla Foods Innovation Centre, Agro Food Park 19, 8200 Aarhus, Denmark; varau@arlafoods.com

**Keywords:** dairy, multiple reaction monitoring, processing, glycation, lysinoalanine, lanthionine

## Abstract

Ultra-high temperature (UHT) processing of milk can result in protein changes during storage; however, the progress of dehydroalanine (DHA) mediated protein cross-linking and Maillard reactions in relation to the sediment formation have not been investigated previously. Liquid chromatography–mass spectrometry, based on multiple reaction monitoring (MRM), was used to absolutely quantify concentrations of furosine, N-ε-(carboxyethyl)lysine (CEL), N-ε-(carboxymethyl)lysine (CML), lanthionine (LAN) and lysinoalanine (LAL) in skim milk and sediment of UHT milk produced from raw milk with either small or large casein micelles. The results showed a higher molar proportion of the advanced stage Maillard reaction products CEL and CML in the sediment, compared to early stage Maillard reaction product furosine, whereas furosine was predominant in the skim milk. Both LAL and LAN increased during storage in the skim milk phase, however only LAL was identified in the sediment. The milk pool with large native casein micelles, known to have a higher percentage of sedimentation, contained higher proportions of furosine, CEL, CML and LAL in the sediment compared to milk with smaller native casein micelles. The study demonstrates the potential contribution of processing-induced protein-protein interactions to sedimentation in UHT milk during storage.

## 1. Introduction

Ultra-high temperature (UHT) processing of milk is utilized to ensure product safety during long shelf-life storage. However, the severe heat treatment induces unwanted changes to the dairy matrix and its components, which can produce molecular processing-induced changes in the proteins and affect their physiochemical properties. These protein modifications occur upon heat treatment; however, their severity extends during storage. The induced protein changes can influence the nutritional value of the products, as well as lead to unwanted off-flavors and color changes, and can potentially contribute to sedimentation or age gelation [1]. Age gelation is the formation of an irreversible three-dimensional protein network induced by proteolytic enzymes (either endogenous or exogenous) and/or physicochemical changes occurring during storage. On the other hand, sedimentation is the formation of a compact protein-rich layer at the bottom of the product. Depending on fat content, fat aggregation can also occur in the top layer. These phenomena have been well-documented as occurring during the storage of UHT milk, but the underlying molecular mechanisms are still not fully understood. Recently, it was found that UHT milk prepared from raw milk containing large casein micelles generated more sediment during a 6-month storage period at 40 °C than UHT milk prepared from milk with smaller casein micelles or stored at a lower temperature [2,3]. It was suggested that rearrangements involving dissociation of β-lactoglobulin-κ-casein complexes from the micelles in relation to UHT treatment and storage could be one driving force, leading to sedimentation of the shaved micelles no longer repulsed by protruding, negative κ-casein tails [3,4]. In addition, proteolysis resulting in κ-casein liberation from the casein micelle was also thought to play a role in sediment formation [3]. However, these studies also suggested that factors other than protein degradation might be involved. Differences in sedimentation between milk pools from raw milk with different casein micelle sizes could indicate an influence of casein micelle density on sedimentation, while a higher sedimentation at high storage temperatures might be due to an increase of protein modifications due to cross-linking of proteins and Maillard reaction [2]. Hence, it was suggested that further study of protein modifications was necessary, in order to fully understand the differences in sedimentation between small and large native casein micelles [2,3].

There are currently two main pathways identified as causing formation of protein cross-linking in processed milk: the Maillard reaction and the dehydroalanine (DHA) pathway. The Maillard reaction is a complex chain of reactions that is initiated between an amine group on a protein/peptide and a reducing sugar, resulting in the formation of different aroma compounds, advanced glycation end products (AGEs) and a large group of cross-linked products, known as melanoidins [5]. In the early stage of Maillard reaction, a condensation reaction is initiated between a carbonyl group in the open chain form of a sugar and a nucleophilic amino group (lysine residues or free N-terminals), resulting in the formation of a Schiff-base, which then rearranges into the more stable Amadori product. In the advanced stage of the Maillard reaction, the Amadori product can be degraded to a reactive α-dicarbonyl compound, which can later react with free amino acids to form Strecker aldehydes, or with nucleophilic side chains of amino acids bound in peptides or proteins to form different AGEs. Major AGEs previously identified in UHT milk are N-ε-(carboxyethyl)lysine (CEL) and N-ε-(carboxymethyl)lysine (CML) which are peptide- or protein-bound lysine modifications derived from the reaction of the α-dicarbonyls glyoxal and methylglyoxal with a single lysine residue [6]. The reaction of glyoxal and methylglyoxal with two lysine residues bound in the sequence of peptides or proteins will form the inter or intra-protein cross-links, known as glyoxal-lysine dimers (GOLD) and methylglyoxal-lysine dimers (MOLD). Instead of degradation, compounds derived from the Amadori product can suffer cyclation and enolization, forming β-pyranone and 3-furanone. Further isomerization of β-pyranone, and its reaction with proteins, is speculated to be an alternative protein cross-linking formation mechanism [7,8].

On the other hand, the sugar-independent DHA pathway can lead to aggregate formation by covalent cross-linking between specific residues. During the initial stage, either serine, phosphoserine, glycoserine, cysteine or cystine residues within protein chains can undergo a heat- or alkali-induced β-elimination reaction to form the unstable DHA compound. After rearrangement, the alkene of DHA can further react with lysine, cysteine or histidine interchain or intrachain amino acid residues to form covalent cross-links known as lysinoalanine (LAL), lanthionine (LAN) or histidinoalanine (HAL), respectively [9]. Both Maillard reaction and DHA-driven protein cross-linking are known to increase with the intensity of the heat treatment [10], and during storage [5,6].

New methods for quantification of processing-induced markers have been developed in recent years. The use of a liquid chromatography triple quadrupole based method (LC-MS triple Q) with multiple reaction monitoring (MRM) has proven to be a fast and reliable method for LAL and LAN quantification [11]. The prospect of using this novel, quantitative method for simultaneous quantification of multiple processing-induced markers is a new milestone in the study of protein cross-linking in foods. Recently, targeted MS analysis has been successfully used in the study of lactose influence on Maillard reaction and DHA-mediated protein cross-linking in caseins and whey proteins [10]. It has also been used to study these processing-induced changes in different food products and plasma [12]. This method enables simultaneous quantification of the development of both sugar-dependent and sugar-independent covalent processing-induced markers during storage of UHT-treated milk. This information is key to the research of the main mechanisms driving aggregate formation and sedimentation during milk storage.

In the present study, it is hypothesized that processing-induced markers are higher in milk with increased levels of sedimentation. To investigate this, samples from a previously conducted storage study [2,3] were analyzed by the MRM-based method for absolute quantification of processing-induced protein modifications.

## 2. Materials and Methods

### 2.1. Formation of Milk Pools, Processing and Storage

Milk collection and processing were performed and described previously by Akkerman, et al., [2]. Based exclusively on casein micelle size from an initial screening of milk from individual cows, two pools of milk, representing either small (82.56 ± 5.54 nm) or large (139.80 ± 4.55 nm) native casein micelles, were formed. Size analysis was performed by dynamic light scattering with 180° heterodyne detection using NANO-flex^®^ (Microtrac, Montgomeryville, PA, USA), and the average size was calculated based on numeric distribution. After collection and pooling of the milk, it was transported to Arla Foods Innovation Center (Aarhus N, Denmark) and stored at 4 °C until further processing. On the following day, the pooled, un-standardized milk samples were homogenized by upstream homogenization at 100 bar and subsequently heat-treated by indirect tubular heat exchange at 141 °C, with a holding time of 4 s, and directly cooled to 4 °C before being tapped into 500 mL pre-sterilized screw-cap plastic bottles, as previously outlined [2]. The heat-treated milk was transported to the laboratories at Aarhus University, and stored at a constant temperature of 40 °C for up to 6 months. Milk and sediment samples were collected in duplicate from separate containers every month. The drawn milk was skimmed by centrifugation at 2643× *g* for 30 min at 4 °C at each time point. Sediment samples were re-dissolved in non-reducing Laemmli gel sample buffer (mM Tris, 2% SDS and 20% glycerol in the ratio 1:99 (*w*/*v*)), and rotated for 8 h to achieve a nearly complete dissolution of the sediment. The skim milk fractions reported here therefore represent the remaining soluble skim milk after sedimentation was allowed to take place and to aggregate at the bottom of the container. Both skim milk and sediment samples were then stored at −20 °C until further analysis. In total, 32 skim milk samples were collected from the two milk pools, representing samples taken before and directly after UHT treatment, and from the six sampling time points (1–6 months), and in duplicates [3]. Only 24 sediment samples were collected, since sediment formation did not occur until the first month of storage. The protein contents were measured by infrared spectroscopy (Milkoscan^®^ FT2, Foss Analytical, Hillerød, Denmark) for the skim milk, and by Bradford assay for re-dissolved sediments (Appendix A).

### 2.2. MRM Quantification Using LC-MS Triple Q

Sample preparation was done according to Nielsen, et al., [11] by mixing 100 µL of skim milk or re-dissolved sediment with 200 µL of 10 M HCl. Nitrogen was flown into the samples for oxygen depletion, and the samples were then heated at 110 °C for 24 h for acid hydrolysis. The hydrolyzed samples were then diluted with 700 µL of water (LC-MS grade), and centrifuged at 14,000× *g* for 15 min. 400 µL of supernatant was collected and dried by vacuum centrifugation (SP Scientific, Stone Ridge, NY, USA) at 40 °C for 1 h and 30 min. Samples were re-dissolved in 400 µL of water (LC-MS grade), filtered (Whatman filters, 0.2 µm) and stored at −20 °C until analysis using LC-MS triple Q, as previously described [10]. In this method, furosine, CEL and CML were used as processing-induced markers representing the Maillard reaction, while LAN and LAL represented processing-induced markers of the DHA cross-linking pathway. Lysine was also simultaneously quantified, as it is a substrate of both pathways. The absolute quantities of the processing-induced markers obtained from the MRM analysis were adjusted to the protein concentration measured in the skim milk and sediment samples. The quantity of each processing marker was converted from the initial measured μg/mL to mol of compound per mol of protein, using the molecular weight of each compound (furosine = 254.28 g/mol, CEL = 218.25 g/mol, CML = 204.1 g/mol, LAL = 233.27 g/mol, LAN = 208.2 g/mol and lysine = 146.19 g/mol). The mol of protein in milk and sediment was calculated based on the mol/L of the four caseins, β-lactoglobulin and α-lactalbumin, in ratios equal to that in milk. The protein composition of the sediment was assumed to be the same as in milk, since individual protein ratios in sediment were not available.

### 2.3. Color Measurement

At each sampling point, color was measured on 10 mL of skimmed milk in a ceramic vessel using a Chroma-meter (CR-400, Konica Minolta Inc., Tokyo, Japan) as described by Akkerman, et al., [2]. The color space was applied according to the CIELAB system [13]: L*, black-white scale, where L* = 0 equals black and L* = 100 equals white; a*, red-green scale, with negative a* = green and positive a* = red; and b*, yellow-blue scale, with negative b* = blue and positive b* = yellow.

### 2.4. Statistical Analysis

Statistical analysis was conducted using the Analysis ToolPak from Microsoft Excel. The two-way analysis of variance test (ANOVA) was performed to test for differences between milk pools (small and large casein size milk) and storage time after UHT processing (0 to 6 months). Differences were considered statistically significant when *p* ≤ 0.05. In sediment, CEL, CML and LAN were not quantified in all samples, and were therefore not included in the statistical model.

## 3. Results

### 3.1. Quantification of Processing-Induced Markers for Maillard Reaction in UHT Skim Milk and Sediment Fractions

The progression in furosine, CEL and CML molar levels in skim milk over the six months of storage, as determined by targeted MS using the MRM method, is shown in Figure 1. Furosine was detected in raw milk in very small amounts, but increased drastically after the UHT treatment. Furosine further significantly increased in the UHT skim milk during storage (*p* < 0.001, Table 1). The molar level of CML and CEL in skim milk was much lower than that of furosine, with CEL having the lowest level. No CML was detected prior to UHT treatment; however, similarly to furosine, the levels of CML and CEL increased significantly in skim milk during storage (*p* < 0.001, Table 1). No statistically significant difference in the molar levels of furosine or CML between skim milk representing small or large casein micelles was observed, while CEL in skim milk was found to differ between milk pools (*p* < 0.01, Table 1). The molar level of CEL was, on average, higher for milk representing small casein micelles compared to milk representing large casein micelles in the UHT skim milk.

Maillard reaction related markers were likewise determined in the isolated and redissolved sediment formed during storage. Prior to and immediately after UHT treatment (time 0, prior to storage) no sediment had formed, and therefore is not present in Figure 1. Furosine content was found to increase significantly during storage (*p* < 0.001, Table 1), and differed between milk pools, relative to the casein micelle size from which the UHT milk was prepared (*p* < 0.001, Table 1), with furosine content being higher in sediment generated from milk with large casein micelles at all storage times. In general, the furosine content was lower in the sediment compared to the skim milk. The level of CEL in sediment from milk with large casein micelles was quite constant throughout storage, while CEL levels in sediment from milk with small casein micelles were below the quantification limit at all storage time points. After three months of storage, CML was detected in the sediment from milk with large casein micelles, and in increasing levels onwards, while for sediment from milk with small casein micelles, CML was detected only at storage months five and six. It is notable that the concentration of CML in the sediment increased beyond that of furosine from storage month three and onwards, for milk with both large and small casein micelles. CEL was detected only in the sediment fraction representing large casein micelles, but its concentration was, on average, around 10-fold higher than the CEL content measured in the skim milk.

### 3.2. Quantification of Processing-Induced Markers for DHA-Mediated Cross-Linking 

The development of the sugar-independent DHA-mediated processing markers, LAL and LAN, in UHT-treated skim milk during storage is shown in Figure 2. Both LAL and LAN, which are the products of cross-linking between DHA and lysine or cysteine residues, respectively, increased significantly during storage (*p* < 0.001, Table 1). In addition, the levels of both LAL and LAN were significantly different in the UHT skim milk prepared from raw milk with large and with small casein micelles, respectively. Milk with large casein micelles had the highest concentration of LAL (*p* < 0.001, Table 1), with the exception of month one, whereas the concentration of LAN was higher in milk with small casein micelles compared to milk with large casein micelles (*p*< 0.001, Table 1). Of these two DHA pathway related markers, LAL was much more abundant than LAN in skim milk. No LAL or LAN was detected prior to UHT treatment.

The two DHA-mediated cross-linking markers were also investigated in the sediment fraction, and the results are shown in Figure 2. Although LAN was detectable from months three to six, it remained below the limit of quantification, and therefore is not included in Figure 2. The molar concentration of LAL increased significantly over time during the storage experiment (*p* < 0.001, Table 1), and a significant difference was found between the LAL content of milk with large and small casein micelles in the sediment (*p* < 0.001, Table 1). On average, the LAL content was approximately 25% higher in sediment from milk with large casein micelles. The level of LAL in the sediment was comparable or higher to the level measured in the skim milk fraction.

### 3.3. Quantification of Total Lysine

Lysine is a substrate for both Maillard reaction and LAL formation, and hence its quantification can be considered as an indirect marker for the level of processing-induced protein changes. The lysine level in skim milk (Figure 3) was reduced after UHT treatment, but otherwise did not change significantly during storage. No significant difference was observed in the lysine content in skim milk between milk pools (large vs small casein micelles). On average, in skim milk, the level of lysine was approximately 11 mol per mol protein, and much higher than in sediment. Despite some increase, the level of lysine in the sediment (Figure 3) similarly showed no overall significant change during the storage time. However, a significant difference in lysine content was observed between sediment formed during storage of UHT milk prepared from the two milk pools, with sediment from milk with small casein micelles having the largest quantity of intact lysine (*p* < 0.05, Table 1).

### 3.4. Color Development in UHT Skim Milk RELATIVE to Furosine Levels during Storage

Color development of the skim milk samples across the storage period was determined by and related to the Maillard reaction. Amongst all the CIELAB measured colors, a correlation was only observed between the yellow-blue color scale (b* value) and the level of furosine across skim milk samples and storage times (R^2^ = 0.88, Figure 4). Comparable results were observed for milk with both small and large casein micelles.

## 4. Discussion

A targeted MS analysis for absolute quantification of processing-induced protein modifications was applied to a storage experiment carried out for UHT-treated milk prepared from raw milk pools representing small or large casein micelles. The level of sedimentation was previously found to be higher in UHT milk prepared from raw milk representing large casein micelles, and to increase with storage [2,3]. The present study therefore investigated whether the level of processing-induced markers was higher in milk with increased levels of sedimentation. The quantification of processing-induced protein modifications in mol per mol of protein used in this study allowed for a more precise comparison between the modifications. The use of this unit enabled a fairer comparison, since it standardized the individual molecular weight of each compound, therefore avoiding processing-markers with a high molecular weight (e.g., furosine) to be overrepresented.

It was found that the molar proportion of furosine increased in both skim milk and sediment with storage, and that it differed in sediment relative to milk pool, with higher proportion in sediment made from UHT milk prepared from raw milk with large casein micelles. However, the furosine level at a molar basis relative to moles of protein was much higher in skim milk as compared with sediment. Both CEL and CML increased in skim milk during storage, though CEL levels in skim milk were very low. In sediment, the level of CEL was fairly constant, and higher than that of skim milk. Furthermore, from three to six months of storage, the level of CML in sediment increased, reaching levels twice as high as those observed in skim milk. The levels of both CEL and CML were higher in sediment derived from milk with large casein micelles compared to milk with small casein micelles. As furosine is an early marker for Maillard reaction, it is also possible that the furosine formed had been converted into later Maillard reaction products, as represented by the quantity of CEL and CML. These results indicate that Maillard-related processes, as represented by both CEL and CML, can be either a contributing driving force, through generation of protein-protein cross-links within or between milk proteins, or a consequence of sedimentation and protein up-concentration in the sediment [4]. These two mechanisms cannot be discriminated by the design of the present study.

The molar level of LAL was somewhat higher in the sediment compared to the skim milk, and UHT milk representing milk with larger casein micelles showed a more pronounced degree of LAL formation compared to UHT milk representing milk with smaller casein micelles. Therefore, DHA-mediated cross-linking represented by LAL could, in addition to Maillard reaction, be a potential contributing factor to sediment formation, through inter-protein crosslinking or the formation of LAL could be a direct consequence of the unique characteristics of the sedimentation. LAL was much more pronounced in skim milk compared to LAN, and LAN was only detected below the limit of quantification in sedimentation. As LAN is the cross-linking product of DHA and cysteine, the result may reflect that the sediments are predominantly composed of α_S1_- and β-casein, which do not contain any cysteine residues and only small amounts of whey proteins [3,4,14].

Since Akkerman, et al., [3] observed that milk with large casein micelles produced higher amounts of sediment during storage, a correlation between quantity of processing-induced markers and sediment formation is suggested by the present study. The effect of protein cross-linking in milk spoilage has been previously discussed. Andrews & Cheeseman [15] suggested that cross-linked proteins due to Maillard reactions aggregate into high-molecular-weight protein aggregates, and drive gelation of UHT milk during storage. On the other hand, some studies have shown that the level of reducing sugars has no effect on milk age gelation [1,4]. However, in this study it was found that milk sediment was associated with late-stage Maillard reaction, which is known to be related to cross-linking [7,8]. Additionally, while Maillard reaction has been widely studied and discarded as the solely causative agent of gelation, none of the previously mentioned approaches considered the influence of sugar-independent cross-link products, such as LAL, which is suggested by the present study to play a role as well.

Other factors partly contributing to sedimentation have also previously been discussed [2]. One such account relates to Stokes’ law [16], stating that particles will move according to their size and density over time. However, it has been suggested that this phenomenon may not easily be applied to the complex system of homogenized and UHT-treated milk [2]. Another account related to the dissociation rate of β-lactoglobulin-κ-casein complexes, as the relative distribution of total κ-casein in the untreated milk was not different between milk representing large and small casein micelles [2] which was also observed in other studies [17,18,19]. Presumably, this means that the surface of micelles in milk with large casein micelles are less covered by κ-casein compared to milk with smaller casein micelles. A looser structure of the casein micelles can ease proteolysis and dissociation of the of β-lactoglobulin-κ-casein complexes, and as a result lead to increased level of sediment in milk with large casein micelles. Furthermore plasmin activity in the milk has been related to casein micelle size, which supports a previous study showing that c-terminal-derived κ-casein peptides correlated with the level of sediment in the milk [3]. This indicates that dissociation of κ-casein from the casein micelle may destabilize the casein micelle, causing formation of sediment [3]. The level of sediment could also relate to pH and ionic calcium levels, as a previous study observed that ionic calcium above 1.5 mM and pH below 6.7 resulted in increased levels of sediment. However, a relationship between pH and ionic calcium could not be confirmed in our previous study [2]. Sedimentation in UHT milk is likely induced through the synergy of several mechanisms, where Maillard reaction and DHA-mediated crosslinking could potentially contribute.

The level of lysine, determined by the MRM-based MS method, represented lysine originating from proteins, peptides and free lysine, as the method relies on complete hydrolysis of proteins into amino acid residues and their derivatives using acid hydrolysis. The result showed that the percentage of lysine did not show any significant difference during storage. However, a large difference in concentration between skim milk and sediment was observed, although the molar concentration was adjusted for protein concentration. One explanation could be that a much higher formation of processing-induced protein modifications existed in the sediment. These could include modifications not analyzed in the MRM method applied in this study, such as the late-stage Maillard reaction products GOLD and MOLD, which have previously been identified in UHT milk [6], and which use lysine as a substrate for their reaction. The level of lysine was lower in sediment from milk representing larger casein micelles, and therefore supports the higher level of processing-induced protein modification in milk with larger casein micelles compared with milk with smaller casein milk.

The analytical method used in the present study is highly sensitive. However, there is a limitation to both detection and quantification of the compounds in these samples. Although both CEL and LAN was detected in sediment samples, they were not able to be quantified, as their concentration levels were below the quantitative range of the MS instrument, which was 3.9 ng/mL. Below this, the quantitative accuracy of the analytical standard was outside the defined threshold of 15% [10].

As furosine is a relevant indicator for discerning the onset of Maillard reaction, methods to quickly and reliably measure furosine formation have been widely discussed. Some studies have showcased a potential correlation between furosine and browning, more specifically measurement of b* value [20]. Lysine-derived compounds (such as furosine) have been shown in previous studies to have a higher impact on browning than other amino acid products [21]. In this study, it was possible to corroborate a direct relation between b* value and furosine concentration, especially after the second month of storage. These results are in accordance with those observed in Figure 1, where furosine content was shown to be the main Maillard reaction product in skim milk, especially for the last months of storage, hence being the main driver of browning.

## 5. Conclusions

The results showcased in this article are a new milestone in the fundamental research of protein-protein interaction and its effects on dairy products. For the first time, these processing-induced markers have been analyzed both in milk and sediment at the same time. A higher quantity of AGE was found in the sediment, indicating a preference for late-stage Maillard reactions. At the same time, milk with large casein micelles was found to produce a higher quantity of sediment, and to contain a higher quantity of processing-induced markers in the sediment. While a correlation between quantity of processing-induced markers and sediment formation is clear, the causality is still uncertain. Further research is necessary to elucidate if higher processing-induced modifications are a driving force of sedimentation or if, on the other hand, sedimentation creates an environment that allows processing-induced markers to thrive. Regardless, as found in this article, LAL clearly plays a fundamental role in protein-protein aggregation in milk storage, and should be considered together with Maillard reaction in future research.

## Figures and Tables

**Figure 1 foods-11-01525-f001:**
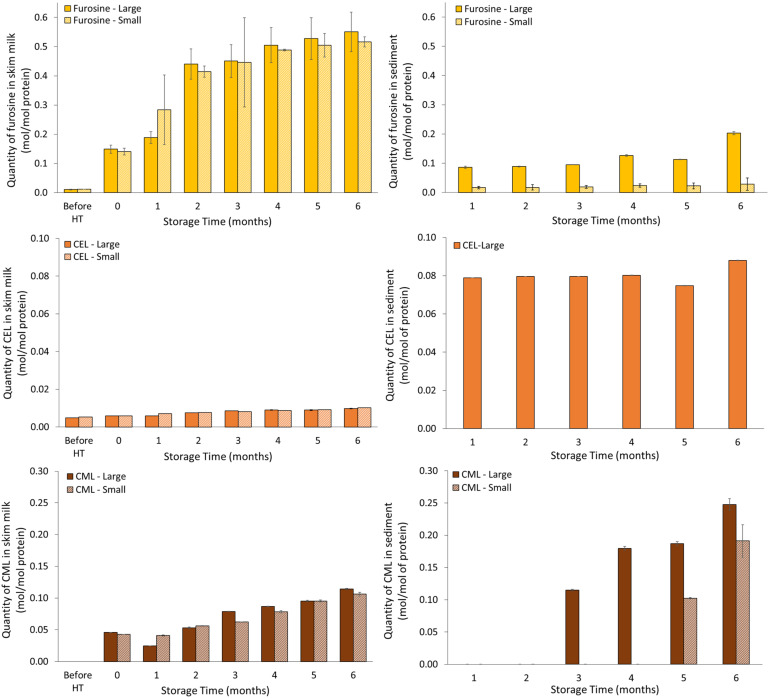
Development in molar levels of furosine, CEL and CML in UHT milk for (**left**) skim milk and (**right**) sediment, representing milk pools with either large or small casein micelles during storage at 40 °C. CEL was below quantification limit on sediment for milk with small casein micelles. ‘Large’ and ‘Small’ indicate UHT milk prepared from raw milk representing large or small casein micelles, respectively.

**Figure 2 foods-11-01525-f002:**
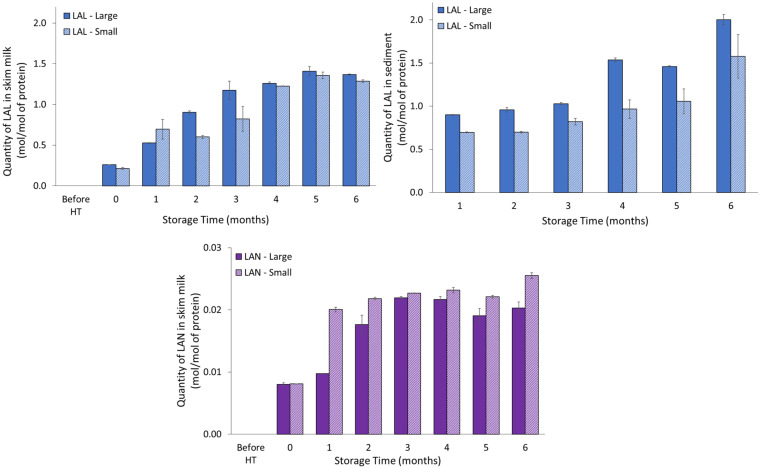
Development in molar levels of LAL and LAN in UHT milk for (**left**) skim milk and (**right**) sediment, and (**below**) LAN in UHT skim milk, representing milk pools with either large or small casein micelles during storage at 40 °C. LAN was below quantification limit in the sediment. ‘Large’ and ‘Small’ indicate UHT milk prepared from raw milk representing large or small casein micelles, respectively.

**Figure 3 foods-11-01525-f003:**
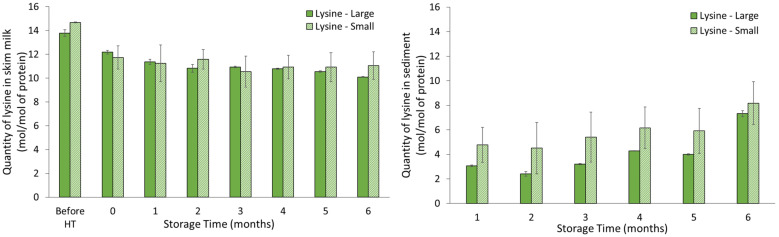
Development in molar levels of lysine quantification in UHT milk for (**left**) skim milk and (**right**) sediment during storage at 40 °C. ‘Large’ and ‘Small’ indicate UHT milk prepared from raw milk representing large or small casein micelles, respectively.

**Figure 4 foods-11-01525-f004:**
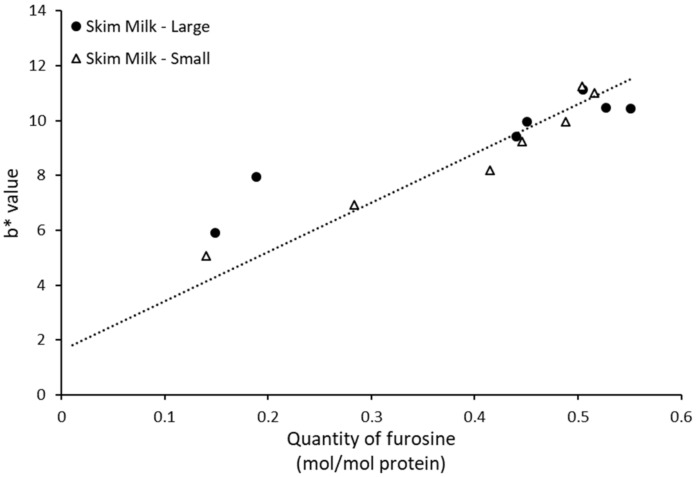
Relationship between b* value (yellow-blue scale, positive values indicates yellowness) and level of furosine in UHT-treated skim milk, representing milk pools with either large or small casein micelles, as developed during the 6-month storage period at 40 °C.

**Table 1 foods-11-01525-t001:** The two-way analysis of variance test (ANOVA) between milk pools (small and large casein size milk) and storage time after UHT processing (0 to 6 months). Differences were considered statistically significant when *p* ≤ 0.05.

Processing-Induced Marker	UHT Milk Fraction	Effect of Milk Pools ^a^	Effect of Storage Time ^a^
**Furosine**	Skim milk	NS	***
	Sediment	***	***
**CEL**	Skim milk	**	***
	Sediment	N/A	N/A
**CML**	Skim milk	NS	***
	Sediment	N/A	N/A
**LAL**	Skim milk	**	***
	Sediment	***	***
**LAN**	Skim milk	***	***
	Sediment	N/A	N/A
**Lysine**	Skim milk	NS	NS
	Sediment	*	NS

^a^ NS, not significant; * *p* < 0.05; ** *p* < 0.01; *** *p* < 0.001; N/A, not applicable.

## Data Availability

Data is contained within the article or Appendix A.

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
