# Peer review of "Development in Maillard Reaction and Dehydroalanine Pathway Markers during Storage of UHT Milk Representing Differences in Casein Micelle Size and Sedimentation"

_foods, 2022, doi:10.3390/foods11101525_

Round 1

Reviewer 1 Report

The authors have taken an original approach to determine Maillard and DHA pathway markers in the sediment of UHT milk with initial differences in casein micelle sizes. The outcomes are valuable to get a better understanding of the age gelation phenomenon in UHT milk. 

I have a few comments:
The paper uses the same samples as previous work of Akkerman et al. (ref 2,3). It refers to these papers for general composition of the sediment and some other aspects (e.g. line 265, line 371). It would be useful to repeat sediment and protein related values found in that paper here, either in text or in the supplement, which will aid to the understanding of the results without making the reader look-up the other papers.

Considering the importance of protein interactions, it would be usefull to know the protein content of the sediment as well, was this determined? In line 269 the authors state that quantification in mol/mol protein in milk allowed more precise comparison. Please explain why. Why did you not also express as mol/mol protein in sediment?

line 51; can you be more specific which results indicated that other factors might be involved? This seems essential for this study

line 105: state the method used for size analysis and the type of size distribution used to determine the averages

Author Response

Dear reviewer, 

Thank you kindly for your inputs and suggestions. Here are the changes that we have done accordingly:

  • Regarding the inclusion of results from previous papers (Akkerman et al.) we have decided to minimize it's use in order to avoid duplicating results on two different papers, so for now he have decided to reference the results to those papers.
  • We have, however, included a supplementary table with the protein content of the sediment samples as suggested. Additionally we have made some clarifications in line 155 to show that the mols per protein were calculated for both skim milk and sediment, using different protein quantity but the same protein ratio.
  • In line 286, additional information has been added regarding the benefits of using mol/mol of protein as a unit.
  • In line 51, we have modify the text to include more information on the reasoning behind the suggestion of previous authors to study protein modifications.
  • In line 114, method and type of size distribution has been added.

Attached you can find the newest version of the paper with all the changes from the last version tracked.

Kind regards,

Miguel

Reviewer 2 Report

The article is quite interesting and well written. I have no substantive comments.
I only suggest adding a list of abbreviations and moving table S1 to the results chapter (Why Table S1, not Table 1?). Figure 1 might be clearer if furosine, CEL, CML were presented separately, as well as LAL and LAN in figure 2. The above remarks do not influence the high substantive value of this work.

Author Response

Dear reviewer, 

Thank you kindly for your inputs and suggestions. Here are the changes that we have done accordingly:

  • We are not sure of the possibility to use an abbreviation list within the Foods format, so we decided to mention the meaning of each abbreviation the first time that they are used. 
  • We included Table S1 in the paper and now is Table 1. 
  • We separated each processing-induced marker on the figures as suggested to improve clarity.

Attached you can find the newest version of the paper with all the changes from the last version tracked.

Kind regards,

Miguel
